# Rising Lysophosphatidylcholine Levels Post-Hepatitis C Clearance

**DOI:** 10.3390/ijms25021198

**Published:** 2024-01-18

**Authors:** Georg Peschel, Sabrina Krautbauer, Kilian Weigand, Jonathan Grimm, Marcus Höring, Gerhard Liebisch, Martina Müller, Christa Buechler

**Affiliations:** 1Department of Internal Medicine I, Gastroenterology, Hepatology, Endocrinology, Rheumatology, and Infectious Diseases, University Hospital Regensburg, 93053 Regensburg, Germany; georg.peschel@klinikum-ffb.de (G.P.); kilian.weigand@gk.de (K.W.); jonathan.grimm@stud.uni-regensburg.de (J.G.); martina.mueller-schilling@klinik.uni-regensburg.de (M.M.); 2Department of Internal Medicine, Klinikum Fürstenfeldbruck, 82256 Fürstenfeldbruck, Germany; 3Institute of Clinical Chemistry and Laboratory Medicine, University Hospital Regensburg, 93053 Regensburg, Germany; sabrina.krautbauer@klinik.uni-regensburg.de (S.K.); marcus.hoering@klinik.uni-regensbug.de (M.H.); gerhard.liebisch@klinik.uni-regensbug.de (G.L.); 4Department of Gastroenterology, Gemeinschaftsklinikum Mittelrhein, 56073 Koblenz, Germany

**Keywords:** HCV, direct-acting antivirals, model of end-stage liver disease score, liver cirrhosis, receiver operating characteristic curve, lysophosphatidylcholine

## Abstract

Hepatitis C virus (HCV) infection alters lysophosphatidylcholine (LPC) metabolism, enhancing viral infectivity and replication. Direct-acting antivirals (DAAs) effectively treat HCV and rapidly normalize serum cholesterol. In serum, LPC species are primarily albumin-bound but are also present in lipoprotein particles. This study aims to assess the impact of HCV eradication on serum LPC species levels in patients infected with HCV. Therefore, 12 different LPC species were measured by electrospray ionization tandem mass spectrometry (ESI-MS/MS) in the sera of 178 patients with chronic HCV infections at baseline, and in 176 of these patients after therapy with DAAs. All LPC species increased at 4 and 12 weeks post-initiation of DAA therapy. The serum profiles of the LPC species were similar before and after the viral cure. Patients with HCV and liver cirrhosis exhibited lower serum levels of all LPC species, except LPC 16:1, both before and after DAA treatment. Percentages of LPC 18:1 (relative to the total LPC level) were higher, and % LPC 22:5 and 22:6 were lower in cirrhotic compared to non-cirrhotic patients at baseline and at the end of therapy. LPC species levels inversely correlated with the model of end-stage liver disease score and directly with baseline and post-therapy albumin levels. Receiver operating characteristic curve analysis indicated an area under the curve of 0.773 and 0.720 for % LPC 18:1 (relative to total LPC levels) for classifying fibrosis at baseline and post-therapy, respectively. In summary, HCV elimination was found to increase all LPC species and elevated LPC 18:1 relative to total LPC levels may have pathological significance in HCV-related liver cirrhosis.

## 1. Introduction

Chronic infection with hepatitis C virus (HCV) is a leading cause of liver diseases, which can progress to liver fibrosis and liver cirrhosis [1]. HCV infection is tightly linked with patients’ lipid metabolisms, and the virus relies on the hosts’ lipoprotein machineries for most of its life cycle [2,3]. 

Direct-acting antivirals (DAAs) eradicate HCV within weeks, achieving up to a 100% sustained virological response (SVR) [4]. Serum total cholesterol and low density lipoprotein (LDL) levels are reduced in HCV infection, and normalize early during DAA treatment [5,6,7,8]. LDL carries different lipid classes such as cholesteryl esters, phosphatidylcholines and lysophosphatidylcholines (LPCs), and LPCs make up about 1% of LDL lipids [9]. Most serum LPC lipids are bound to albumin [9,10]. 

Currently, the effect of DAA therapy on circulating LPC levels remains unclear. Enzymes that use LPC as a substrate influence HCV infectivity and replication [11,12], suggesting that serum LPC analysis may provide insight into HCV pathology.

LPC acyltransferase 1 (LPCAT1) converts LPC to phosphatidylcholine, and is downregulated in HCV-infected hepatocytes [11]. The knockdown of LPCAT1 increased the production of highly infectious lipid-enriched HCV particles, confirming a role for LPC in HCV severity [11].

Autotaxin is a circulating enzyme that converts LPC to lysophosphatidic acid [13]. Circulating autotaxin levels were elevated in HCV-infected patients [14,15], and DAA therapy caused a corresponding decrease in serum autotaxin levels [16,17]. 

Autotaxin is, moreover, a biomarker for liver fibrosis [16,17]. Serum autotaxin levels positively correlated with liver fibrosis stages of HCV patients before therapy and at SVR [16]. 

Cirrhosis of the liver is characterized by low serum levels of LDL, high-density lipoprotein (HDL) and albumin [18,19,20,21], and lipids carried in LDL and HDL are also reduced [9,22].

In HCV patients, advanced liver cirrhosis was associated with lower plasma levels of LPC 16:0, 18:0, 20:3, 20:4 and 22:6 [23]. A second study could not observe a decline in LPC species in patients with HCV-induced liver cirrhosis. This analysis showed that LPC 14:0, 16:0, 18:3, 18:0, 20:5, 20:4, 20:3, 22:5 and 22:6 levels were already low in HCV infection, in contrast to healthy controls. In this patient cohort, LPC 14:0, 16:1, 18:1 and 20:1 were higher in patients with HCV cirrhosis compared to patients with chronic HCV without liver cirrhosis [24]. 

LPC species differ in the length of the acyl chain and the number of double bonds, which affects their biological activity [9,25]. LPC 16:0, with levels up to 200 µM in human plasma, is the most abundant species, LPC 18:0 is the second most common species, and LPC 18:1 and 18:2 are the third most abundant species. Circulating levels of other LPC species are comparably low, with plasma concentrations of 10 µM or less [26]. 

However, the biological activities of individual LPC species are largely unknown. 

Our study aimed to measure serum levels of the LPC species 15:0, 16:0, 16:1, 18:0, 18:1, 18:2, 18:3, 20:3, 20:4, 20:5, 22:5 and 22:6 in HCV patients before and after DAA therapy. The goal was to investigate the impact of HCV on circulating LPC levels and identify associations with viral load, viral genotype, and liver disease severity.

## 2. Results

### 2.1. Serum LPC Species in Relation to Age, Gender, Body Mass Index, Liver Steatosis and Diabetes 

Twelve LPC species were determined in sera of 178 patients with chronic HCV. Details of the cohort are listed in Table 1.

Levels of these LPC species did not differ between the 74 females and 104 males (Figure 1a) and did not correlate with the BMI (Appendix A). Age was negatively associated with LPC 16:0, 18:2, 20:3, 20:4, 20:5, 22:5 and 22:6 (Appendix A). The 74 patients with liver steatosis diagnosed by ultrasound imaging had lower levels of serum LPC 15:0 and 16:0 compared to the 104 patients without liver steatosis (Figure 1b). Levels of LPC 16:0, 16:1, 18:0, 20:3, 20:4, 20:5 and 22:6 were reduced in the sera of the 20 patients with diabetes (Figure 1c). Patients with diabetes were more likely to have liver cirrhosis (*p* < 0.001), and lower levels of LPC species may be related to underlying liver injury. 

### 2.2. LPC Species in Relation to Liver Fibrosis

It is well known that patients with advanced liver fibrosis have low serum lipoprotein levels [27]. The 40 HCV patients with ultrasound-diagnosed liver cirrhosis had reduced LPC 15:0, 16:0, 18:0, 18:2, 20:3, 20:4, 20:5, 22:5 and 22:6 levels in serum compared to HCV patients with normal liver function (Figure 2a). 

The fibrosis-4 (FIB-4) score is a serological marker to discriminate patients with and without liver fibrosis [28]. According to the FIB-4 score, patients with significant fibrosis had lower LPC 15:0, 16:0, 18:0, 18:1, 18:2, 18:3, 20:3, 20:4, 20:5, 22:5 and 22:6 levels than patients without fibrosis. LPC 15:0, 16:0, 18:0, 18:2, 20:3, 20:4, 20:5 and 22:6 levels of patients with severe fibrosis were reduced compared to patients with intermediate FIB-4 scores. LPC 20:4 was the only species that differed between patients with no fibrosis and those with an intermediate FIB-4 scores (Figure 2b). 

The LPC species profile of patients with and without ultrasound-diagnosed liver cirrhosis was quite similar. Percent LPC 18:1 (relative to total LPC concentration) was higher, and % LPC 16:0, 20:5 and 22:6 (relative to total LPC concentration) were lower in cirrhosis compared to non-cirrhosis patients (Figure 2c). 

### 2.3. LPC Species in Relation to the Model of End-Stage Liver Disease (MELD) Score and Laboratory Measures of Liver and Kidney Function 

Since most LPC species levels were low in cirrhosis (Figure 2a,b), we calculated correlations of LPC species with laboratory measures separately in patients with and without ultrasound-diagnosed cirrhosis. Correlations with age disappeared when patients were categorized according to liver cirrhosis (Appendix A). Patients with liver cirrhosis were older (*p* < 0.001) and the negative correlations of LPC species with age in the whole cohort are clearly due to the lower LPC levels in cirrhosis patients (Appendix A). 

In patients without liver cirrhosis, total LPC levels negatively correlated with the MELD score (r = −0.291, *p* = 0.007), and this is based on the negative association of LPC 15:0, 16:0, 18:0, 18:2, 20:4 and 20:5 with the MELD score (Table 2). These LPC species and also LPC 20:3 were negatively correlated with the international normalized ratio (INR). All except LPC 16:1, 18:0 and 22:6 negatively correlated with aspartate aminotransferase (AST). LPC 20:3, 20:4 and 20:5 were negatively correlated with alanine aminotransferase (ALT). LPC species did not correlate with albumin, bilirubin or creatinine (Table 2).

In the subgroup of patients with liver cirrhosis, LPC 18:0 and 22:6 were negatively correlated with INR and MELD score. Here, total LPC levels were not related to the MELD score (r = −0.341, *p* = 0.407). Albumin positively correlated with LPC 15:0, 16:0, 18:0, 20:3, 20:4 and 22:6 (Table 3). ALT, AST, bilirubin and creatinine did not correlate with any of the LPC species (Table 3). 

### 2.4. LPC Species in Relation to Laboratory Measures of Inflammation and Platelet Count 

In patients without liver cirrhosis, LPC species did not correlate with C-reactive protein (CRP) or leukocyte count (Appendix A). LPC 18:3 and 20:5 were positively associated with platelet number (Appendix A). 

In the subgroup of patients with liver cirrhosis, none of the LPC species were related with CRP, leukocyte count or platelets (Appendix A).

### 2.5. LPC Species in Relation to Viral Load, Viral Genotype and Viral Cure

None of the LPC species correlated with viral load (Appendix A). In the entire cohort, the 53 patients infected with genotype 1a, the 76 patients infected with genotype 1b, the 35 patients infected with genotype 3a and the 14 patients with less common HCV genotypes, which were grouped together, had similar levels of the LPC species (*p* > 0.05 for all). 

All LPC species were increased compared to pretreatment levels at 4 weeks and 12 weeks after the start of therapy. Higher levels at the end of therapy, relative to pretreatment levels, were observed in cirrhosis and non-cirrhosis patients (Figure 3a). Probably due to the smaller group size of the cirrhosis cohort, this effect was only significant for LPC 15:0 (*p* = 0.007), 16:0 (*p* = 0.047), 16:1 (*p* = 0.002), 18:1 (*p* = 0.029) and 20:5 (*p* = 0.007). 

Consistent with higher levels of all LPC species at the end of DAA therapy, relative levels of LPC species (% of total LPC levels) before therapy and at the end of therapy did not greatly differ. In the non-cirrhosis group, % LPC 18:0 was lower at the end of therapy (Figure 3b). LPC composition in the sera of cirrhosis patients did not significantly change during therapy. 

### 2.6. LPC Species in Relation to the MELD Score and Laboratory Measures Post-DAA Therapy 

At the end of therapy, patients with liver cirrhosis had low levels of all LPC species except LPC 16:1, which was comparable to patients without liver cirrhosis (Appendix A). Percent LPC 18:1 was higher (*p* < 0.001), and % of LPC 20:4 (*p* < 0.001), 20:5 (*p* = 0.010) and 22:6 (*p* < 0.001) were reduced in cirrhosis. 

At the end of treatment, LPC species did not correlate with CRP and leukocyte count in patients with and without ultrasound-diagnosed liver cirrhosis (*p* > 0.05 for all). 

In the non-cirrhosis subgroup, LPC 18:2 (r = 0.282, *p* = 0.013), 18:3 (r = 0.263, *p* = 0.028) and 20:5 (r = 0.339, *p* = 0.001) were related to platelet count. Such associations were not observed in patients with liver cirrhosis. 

In patients without liver cirrhosis, all except LPC 18:2, 18:3, 20:4, 20:5 and 22:6 positively correlated with albumin. Associations of LPC species with the MELD score, INR, ALT, AST, creatinine or bilirubin were not observed (Table 4). 

In patients with ultrasound-diagnosed liver cirrhosis, LPC 16:0, 18:0, 20:4 and 22:6 negatively correlated with the MELD score. LPC 16:0, 18:0 and 22:6 negatively correlated with INR. LPC 15:0, 16:0, 18:0, 20:4 and 22:6 positively correlated with albumin. LPC 15:0 negatively correlated with AST, and LPC 18:0 and 20:4 with bilirubin (Table 5). 

### 2.7. LPC Species for the Discrimination of Patients with and without Liver Cirrhosis

Receiver operating characteristic (ROC) curve analysis was used to identify LPC species that could discriminate patients with and without liver cirrhosis, and the corresponding area under the curve (AUC) values are shown in Table 6. The AUC of the MELD score was 0.898 before and 0.877 after therapy with DAAs, which is considered excellent [29]. An AUC of 0.7 to 0.8 is considered acceptable [29], and an AUC of 0.773 and 0.720 was calculated for % LPC 18:1 before and after therapy, respectively (Table 6). Due to the relatively low AUC of % LPC 18:1 compared to the MELD score, it is not recommended to use % LPC 18:1 as a diagnostic tool. However, it is reasonable to assume that higher levels of LPC 18:1 relative to total serum LPC levels may contribute to the pathogenesis of liver fibrosis and cirrhosis. 

## 3. Discussion

This study demonstrates that LPC species and LDL levels recover simultaneously following HCV elimination. Except for LPC 16:1, all LPC species are reduced in the sera of patients with liver cirrhosis. The serum ratio of LPC 18:1 to total LPC concentrations effectively distinguishes between non-cirrhotic and cirrhotic patients, indicating a potential pathophysiological role of LPC 18:1 in liver cirrhosis.

Chronic liver injury is related to the depletion of circulating LPC levels, and this was reported for hepatitis B virus (HBV) infection, non-alcoholic fatty liver disease and alcoholic hepatitis [30]. The current analysis shows that infection with HCV causes lower serum LPC levels, which recover early after starting DAA therapy. 

Previous studies have consistently found that serum total cholesterol and LDL levels are induced early after the start of DAA treatment [5,6,7,8]. LPC is mainly found in HDL and albumin and about 1% in LDL [9]. The LDL levels of our patients at the end of therapy were 125% of the baseline concentrations [7], and the increase in the different LPC species ranged from 104% to 144%. HDL and albumin levels did not significantly change during DAA treatment, and higher LPC levels in serum at the end of therapy may be associated with the recovery of LDL. It is also possible that HDL composition changes after a viral cure, or that the number of LPC species molecules bound per albumin molecule is increased. 

LPC species levels in serum were not associated with viral load or viral genotype. HCV genotypes 1 and 3 are common genotypes [31], and most of our patients were infected with 1a, 1b and 3a. The genotype 3 infection has been described to affect lipid classes such as ceramides or lathosterol more than the genotype 1 infection [22,32]. LPC levels in the serum of our cohort did not differ between genotypes, suggesting that the LPC composition of lipoprotein particles between HCV genotype 1- and genotype 3-infected patients is similar. 

Autotaxin, which converts LPC to lysophosphatidic acid, is higher in HCV infection and declines at SVR [17]. Lysophosphatidic acid has not been analyzed in the sera of our patients, but higher LPC levels of patients at the end of therapy may be related to a reduced conversion of LPC to lysophosphatidic acid. Phospholipase A2, which converts phosphatidylcholine to LPC, was found to be reduced at SVR [33], indicating that less LPC is provided by this pathway. LCAT produces cholesteryl ester and converts phosphatidylcholine to LPC [34], and this pathway may also contribute to increased serum LPC at the end of therapy. 

Liver cirrhosis is the final stage of chronic liver diseases, irrespective of its etiology [35,36,37]. The liver is the main organ that controls lipid metabolism, and patients with liver cirrhosis have reduced serum lipid and lipoprotein levels [18,22,38]. All but LPC 16:1 were reduced the in sera of patients with liver cirrhosis, regardless of whether liver cirrhosis was diagnosed by ultrasound or by the FIB-4 score. A decline in most LPC species in cirrhosis was observed before therapy and at the end of therapy. Our observation is in accordance with a previous study having reported lower levels of LPC 16:0, 18:0, 20:3, 20:4 and 22:6 with increasing liver injury, assessed by the Child–Turcotte–Pugh score in HCV infection [23]. Patients with HBV cirrhosis had lower serum levels of LPC species in comparison to healthy controls and patients with chronic HBV [39]. Higher fibrosis stages of patients with non-alcoholic fatty liver disease were associated with a decline in LPC 16:0, 18:0, 20:4 and 22:6, whereas LPC 18:1 levels were not altered [40]. Future analysis is required to clarify whether changes in the LPC species in cirrhosis are related to disease etiology. 

Negative correlations of individual LPC species with the MELD score, bilirubin, INR, AST and ALT and positive correlations with albumin were observed in our study. The LPC species, which correlated with these laboratory values, differed between patients with and without cirrhosis and before and after DAA therapy. High correlation coefficients were found for the associations of LPC species with albumin, as would be expected, since most LPC in serum is bound to albumin. 

Diabetes was more common in patients with liver cirrhosis, so the decrease in different LPC species in diabetes is attributed to the underlying liver cirrhosis. LPC 16:1 was reduced in patients with diabetes, but not in those with cirrhosis, suggesting that this is a diabetes-specific change. Associations between LPC species and type 2 diabetes have been described [41,42,43], but a clear relationship between total circulating LPC levels and individual LPC species and diabetes has not been established.

Liver steatosis in our HCV patients was related to lower levels of LPC 16:0. Another study provides evidence for a negative association of LPC species and non-invasive scores for liver fat [44]. In HCV patients, liver steatosis may be caused by viral infection and/or metabolic disease associated with overweight/obesity [45]. In our cohort, LPC species did not correlate with the BMI of the patients and did not differ between patients with a BMI < 25 kg/m^2^ and patients with a BMI > 30 kg/m^2^. This shows that the lower levels of serum LPC 16:0 of patients with liver steatosis are not related to a higher BMI. However, the underlying mechanisms linking specific LPC species to diabetes, steatosis or liver fibrosis are poorly understood. 

Negative correlations of seven LPC species with age identified in the whole cohort did not persist when associations of LPC species with age were analyzed separately in cirrhotic and non-cirrhotic patients. Patients suffering from liver cirrhosis were older, and low levels of LPC species in cirrhosis contributed to negative correlations with age. Thus, age is not a confounding factor for the correlations identified in this study. 

Relative concentrations of LPC 18:1 to total LPC levels in patients with liver cirrhosis were higher before therapy and at the end of therapy. Percentages of LPC 16:0, % LPC 20:5 and % LPC 22:6 were reduced in cirrhosis at baseline and % LPC 20:4, 20:5 and 22:6 at the end of therapy. The pathways contributing to the change in the serum LPC profile and the pathophysiological relevance of an altered LPC species distribution are still unclear. 

ROC curve analysis identified a promising AUC for % LPC 18:1 in differentiating between cirrhotic and non-cirrhotic patients. Although the AUC was not as high as that of the MELD score, making % LPC 18:1 less suitable for the clinical diagnosis of liver cirrhosis, the observed increase in % LPC 18:1 level pre- and post-DAA therapy in liver cirrhosis patients highlights its potential significance in the pathogenesis of liver cirrhosis. Furthermore, our findings provide a basis for future research to explore the role of LPC 18:1 accumulation in liver injury.

## 4. Materials and Methods

### 4.1. Study Cohort

The sera of patients who were not treated for HCV before was collected at the Department of Internal Medicine I (University Hospital of Regensburg) from October 2014 to September 2019 [46]. Venous blood samples were collected in S-Monovette® Serum CAT tubes (Sarstedt, Nürnbrecht, Germany), which contain beads coated with a coagulation activator. Blood clotting occurred within 20 to 30 minutes, and then the samples were centrifuged at 2000 g for 15 min at room temperature. Serum was pipetted into SafeSeal cups (Sarstedt) and immediately stored at −80°C. Patients’ blood was not collected in the fasting state, and non-fasting serum was used for lipid analysis. One study reported that LPC 20:4 and 22:5 become increased in the fasting state, but do not change significantly for up to 6 hours after the ingestion of a test meal [47]. 

All patients were suitable for DAA therapy in accordance with the recommendations of the European Association for the Study of the Liver [4]. Patients were older than 18 years. Exclusion criteria were coinfection with human immunodeficiency virus or HBV, as well as decompensated liver cirrhosis. 

Patients with chronic HCV were treated with one of the following regimens: glecaprevir/pibrentasvir, sofosbuvir/daclatasvir, sofosbuvir/ledipasvir, sofosbuvir/velpatasvir or elbasvir/grazoprevir, following the recommendations of the international guidelines [4]. 

Most of the patients were enrolled early after the approval of interferon-free DAA therapies, and at a time when adverse effects of statins were expected [48], and, therefore, statins were paused during therapy.

Cirrhosis was diagnosed using ultrasound, with criteria including a nodular liver surface, reduced liver size and coarse liver parenchyma [49]. The cut-off values for the FIB-4 scoring were the following. Advanced fibrosis >3.25, no fibrosis: <1.3 for patients younger than 65 years, and <2.0 for patients older than 65 years [50]. 

### 4.2. The Measurement of LPC Species

Serum samples were used for LPC species analysis. Prior to lipid extraction, 25 µL of the internal standard solution, containing LPC 13:0 (0.83 nmol) and LPC 19:0 (0.7 nmol), was placed in a glass centrifuge tube and vacuum-dried. For the calibrators, known amounts of LPC 16:0, LPC 18:0 and LPC 18:1 were added, and the solvent was evaporated. LPC calibrators were 1-acyl-2-hydroxy-*sn*-glycero-3-phosphocholines (Avanti Polar Lipids, Alabaster, AL, USA), and were of >99% purity.

LPC species were analyzed by direct electrospray ionization tandem mass spectrometry (ESI-MS/MS) and this has been described in detail before [51]. A volume of 10 µL serum was extracted according to the procedure described by Bligh and Dyer [52]. The separated chloroform phase was vacuum-dried and dissolved in 7.5 mM ammonium acetate in methanol-chloroform (3:1 by volume). A total of 20 µL of this solution was injected, and data were recorded over 1.3 min. The triple quadrupole mass spectrometer (Quattro LC; Micromass, Waters Cooperation, Milford, MA, USA) was operated in positive SRM, using a product ion of *m*/*z* 184. 

Data analysis was performed using MassLynx software (Version 4.1, Waters GmbH, Eschborn, Germany), which includes the NeoLynx tool for averaging the scans at the half peak height of the total ion count. The NeoLynx results were exported to MS Excel spreadsheets and processed using self-programmed macros. The macros sorted the results, corrected for isotopic overlap (Type II, [53]), calculated ratios to the internal standards, generated calibration curves and calculated quantitative values. Concentrations of the LPC species were determined using the closest related calibration slope. 

### 4.3. The Analysis of Laboratory Parameters and the Calculation of the MELD Score 

Serum creatinine was measured using the Roche Cobas Pro C. The enzymatic method for the determination of serum creatinine was performed by the conversion of creatinine by creatininase, creatinase and sarcosine oxidase to glycine, formaldehyde and hydrogen peroxide. The released hydrogen peroxide is used by peroxidase to form a quinoneiminine dye using 4-aminophenazone and HTIBa as substrates. The color intensity of the quinoneiminine dye is directly proportional to the concentration of creatinine in the reaction mixture.

The GFR was calculated using the equation described by Levey et al. [54]. The Chronic Kidney Disease Epidemiology Collaboration creatinine equation is: GFR = 141 × min (serum creatinine /κ, 1) α × max (serum creatinine /κ, 1) − 1.209 × 0.993age × 1.018 [if female], where κ is 0.7 for females and 0.9 for males and α is −0.329 for females and −0.411 for males [54]. 

CRP was measured using a particle-enhanced immunoturbidimetric assay on the Cobas Pro C analyzer with the appropriate assays (Roche, Penzberg, Germany).

AST catalyzes the transfer of an amino group between L-aspartate and 2-oxoglutarat to form oxalacetate and L-glutamate. Oxalacetate then reacts in the presence of malate dehydrogenase with NADH to form L-malate and NAD+. Pyridoxal phosphate serves as a coenzyme in the aminotransfer reaction. It ensures full enzyme activation. The rate of oxidation of NADH is directly proportional to the catalytic AST activity. It is determined by measuring the extinction rate (Cobas Pro C analyzer).

ALT catalyzes the transfer of an amino group between L-alanine and 2-oxoglutarate to form pyruvate and L-glutamate. Pyruvate then reacts in the presence of lactate dehydrogenase with NADH to form L-lactate and NAD+. Pyridoxal phosphate serves as a coenzyme in the aminotransfer reaction and ensures full enzyme activation. The rate of oxidation of NADH is directly proportional to the catalytic activity of ALT. It is determined by measuring the extinction rate (Cobas Pro C analyzer).

Total bilirubin is coupled with 3,5-dichlorophenyldiazonium in a strongly acidic solution in the presence of a suitable solvent. The color intensity of the red azo dye formed is directly proportional to the total bilirubin, and can be determined photometrically (Cobas Pro C analyzer).

Albumin has sufficient cationic properties at a pH of 4.1 to bind the anionic dye bromocresol green and form a blue–green complex. The intensity of the blue–green color is directly proportional to the albumin concentration in the sample, and is measured photometrically (Cobas Pro C analyzer).

Incubation of citrate-plasma with a standardized amount of thromboplastin and calcium ions initiates the clotting process; the time until the formation of the fibrin core is measured optically. The INR is calculated using the prothrombin time ratio (PR) and the International Sensitivity Index (ISI). PR^ISI^ = INR (BCS XP system, Siemens, Forchheim, Germany).

The analysis for all laboratory parameters mentioned above was performed at the Institute of Clinical Chemistry and Laboratory Medicine at University Hospital Regensburg.

The MELD is a composite score derived from three laboratory values: INR, serum total bilirubin and serum creatinine [55].

### 4.4. Statistical Analysis

The mean values of the LPC species measured are shown as boxes ± standard deviation. Mann–Whitney U-test, ROC analysis, Kruskal–Wallis-test and paired *t*-test were used (IBM SPSS Statistics 26.0 program, SPSS GmbH Software, Muenchen, Germany and MS Office Excel 2016, Microsoft Corporation, Redmond, WA, USA). Data in tables are shown as median, minimum and maximum values. After adjusting for multiple comparisons, a value of *p* < 0.05 was regarded as significant. 

## Figures and Tables

**Figure 1 ijms-25-01198-f001:**
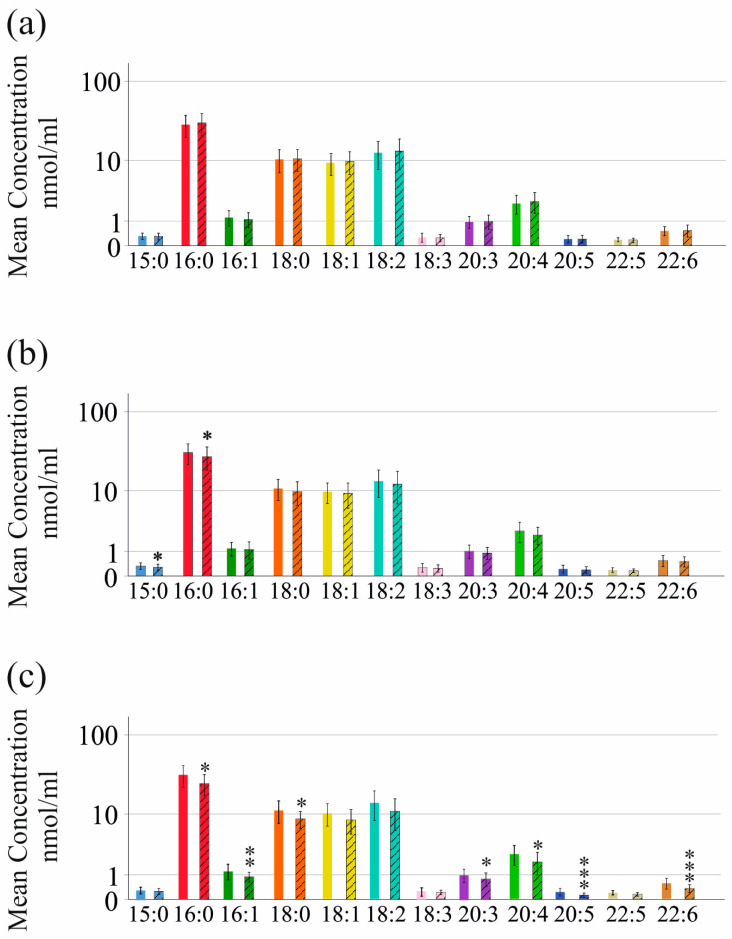
LPC species in sera of HCV patients before therapy. (**a**) LPC species in sera of females and males (hatched bars); (**b**) LPC species in sera of HCV patients without and with liver steatosis (hatched bars); (**c**) LPC species in sera of HCV patients without and with diabetes (hatched bars). Data are shown in a logarithmic scale to improve the visualization of low-abundance LPC species. * *p* < 0.05, ** *p* < 0.01, *** *p* < 0.001. Statistical test used: Mann–Whitney U-test.

**Figure 2 ijms-25-01198-f002:**
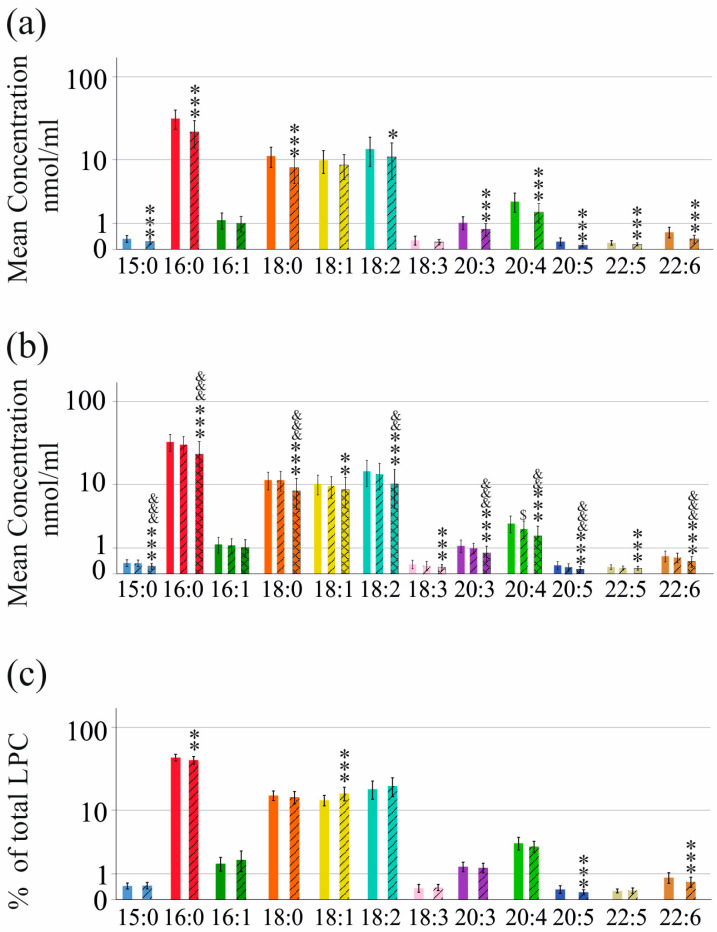
LPC species in sera of HCV patients before therapy in relation to liver cirrhosis. (**a**) LPC species in sera of patients without and with (hatched bars) liver cirrhosis diagnosed by ultrasound; (**b**) LPC species in sera of HCV patients with low, intermediate (hatched bars) and high (zigzag bars) FIB-4 scores; (**c**) LPC species relative to total LPC levels in % in sera of patients without and with (hatched bars) liver cirrhosis diagnosed by ultrasound. Data are shown in a logarithmic scale to improve the visualization of low-abundance LPC species. * *p* < 0.05, ** *p* < 0.01, *** *p* < 0.001 for comparison of patients without cirrhosis/with a low FIB-4 score and patients with cirrhosis/with a high FIB-4 score; ^&&^ *p* < 0.01, ^&&&^ *p* < 0.001 for comparison of patients with an intermediate and a high FIB-4 score; ^$^
*p* < 0.05 for comparison of patients with a low and patients with an intermediate FIB-4 score. Statistical tests used: Mann–Whitney U-test (**a**,**c**) and Kruskal–Wallis test (**b**).

**Figure 3 ijms-25-01198-f003:**
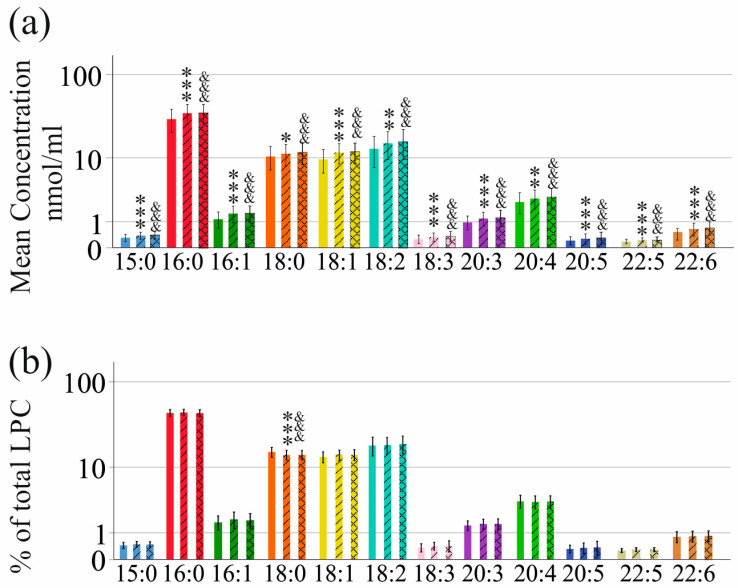
LPC species in sera of HCV patients without liver cirrhosis during DAA therapy. (**a**) LPC species in sera of patients before therapy, at 4 weeks (hatched bars) and 12 weeks (zigzag bars) after start of therapy. (**b**) Serum LPC species levels relative to total LPC concentrations (% of the LPC species relative to total LPC levels) in sera of patients before therapy, at 4 weeks (hatched bars) and 12 weeks (zigzag bars) after start of therapy. Data are shown in a logarithmic scale to improve the visualization of low-abundance LPC species. * *p* < 0.05, ** *p* < 0.01, *** *p* < 0.001 for comparison of baseline levels and LPC levels at 4 weeks after therapy start. ^&&&^ *p* < 0.001 for comparison of baseline levels and LPC levels at 12 weeks after the start of therapy. Statistical test used: paired *t*-test.

**Table 1 ijms-25-01198-t001:** Laboratory parameters of the patients before and after antiviral therapy (alanine aminotransferase (ALT), aspartate aminotransferase (AST), body mass index (BMI), C-reactive protein (CRP), high-density lipoprotein (HDL), international normalized ratio (INR), low-density lipoprotein (LDL), model of end-stage liver disease (MELD), not significant (ns)). Statistical test used: paired *t*-test.

Laboratory Parameter	Baseline (178 Patients)	12 Weeks Therapy (176 Patients)	*p*-Value
Age years	54 (24–82)	54 (24–82)	ns
Female/Male	74/104	74/102	ns
BMI kg/m^2^	25.6 (17.6–41.6)	25.6 (17.6–41.6)	ns
MELD Score	7 (6–21)	7 (6–21)	ns
Ferritin ng/mL	128.6 (5.6–2309)	94.2 (2.9–1161)	0.003
ALT U/L	61 (2–305)	26 (6–388)	<0.001
AST U/L	47 (7–1230)	22 (6–836)	<0.001
Bilirubin mg/dL	1.0 (1.0–4.3)	1.0 (1.0–7.5)	ns
INR	1.05 (1.00–2.44)	1.04 (1.00–2.22)	ns
Creatinine mg/dL	0.78 (0.14–14.00)	0.76 (0.14–14.7)	ns
Platelets n/nL	195 (38–402)	206 (37–407)	ns
Leukocytes n/L	6.5 (2.2–72.4)	6.8 (2.4–62.9)	ns
CRP mg/L	2.9 (1.0–55.0)	2.9 (2.9–20.3)	ns
Albumin g/L	38 (2–50)	39 (16–93)	ns
HDL mg/dL	52 (19–111)	50 (13–96)	ns
LDL mg/dL	95 (23–296)	119 (33–251)	<0.001

**Table 2 ijms-25-01198-t002:** Spearman correlation coefficients for the correlation of LPC species and the MELD score as well as laboratory measures of patients without liver cirrhosis before therapy (alanine aminotransferase (ALT), aspartate aminotransferase (AST), international normalized ratio (INR), model of end-stage liver disease (MELD)). * *p* < 0.05. ** *p* < 0.01, *** *p* < 0.001.

LPC nmol/mL	MELD Score	INR	Albumin g/L	ALT U/L	AST U/L	Creatinine mg/dL	Bilirubin mg/dL
15:0	−0.263 *	−0.262 *	0.147	0.217	−0.322 **	0.048	−0.046
16:0	−0.284 *	−0.305 **	0.211	−0.156	−0.266 *	0.110	−0.041
16:1	−0.161	−0.124	0.079	−0.198	−0.187	−0.028	−0.048
18:0	−0.282 *	−0.302 **	0.215	−0.117	−0.199	0.068	−0.017
18:1	−0.203	−0.216	0.197	−0.226	−0.304 **	0.017	−0.010
18:2	−0.245 *	−0.275 *	0.269	−0.243	−0.336 **	0.035	−0.081
18:3	−0.212	−0.193	0.091	−0.229	−0.292 **	0.072	−0.133
20:3	−0.241	−0.263 *	0.072	−0.248 *	−0.352 ***	0.096	−0.064
20:4	−0.246 *	−0.276 *	0.131	−0.251 *	−0.400 ***	0.188	−0.017
20:5	−0.279 *	−0.251 *	0.054	−0.281 *	−0.415 ***	0.114	−0.014
22:5	−0.127	−0.085	0.057	−0.181	−0.310 **	0.128	−0.003
22:6	−0.186	−0.193	0.137	−0.140	−0.237	0.068	0.086

**Table 3 ijms-25-01198-t003:** Spearman correlation coefficients for the correlation of LPC species and the MELD score as well as laboratory measures of patients with liver cirrhosis before therapy (alanine aminotransferase (ALT), aspartate aminotransferase (AST), international normalized ratio (INR), model of end-stage liver disease (MELD)). * *p* < 0.05. ** *p* < 0.01, *** *p* < 0.001.

LPC nmol/mL	MELD Score	INR	Albumin g/L	ALT U/L	AST U/L	Creatinine mg/dL	Bilirubin mg/dL
15:0	−0.187	−0.288	0.655 ***	−0.003	−0.392	0.079	−0.121
16:0	−0.346	−0.375	0.688 ***	−0.141	−0.446	−0.066	−0.297
16:1	−0.044	0.001	0.293	−0.244	−0.311	−0.086	0.076
18:0	−0.455 *	−0.490 *	0.731 ***	−0.024	−0.341	−0.051	−0.419
18:1	−0.072	−0.003	0.260	−0.307	−0.357	−0.210	−0.020
18:2	−0.224	−0.069	0.380	−0.111	−0.299	−0.156	−0.217
18:3	−0.044	0.122	0.194	−0.194	−0.306	−0.218	−0.100
20:3	−0.307	−0.278	0.459 *	−0.082	−0.388	−0.162	−0.276
20:4	−0.353	−0.314	0.628 ***	−0.150	−0.440	−0.072	−0.293
20:5	−0.195	−0.226	0.349	−0.169	−0.425	−0.081	−0.126
22:5	−0.247	−0.201	0.328	−0.134	−0.210	−0.288	−0.043
22:6	−0.549 **	−0.556 **	0.717 ***	0.004	−0.331	−0.020	−0.415

**Table 4 ijms-25-01198-t004:** Spearman correlation coefficients for the correlation of LPC species and the MELD score, as well as laboratory measures of patients without liver cirrhosis at the end of therapy (alanine aminotransferase (ALT), aspartate aminotransferase (AST), international normalized ratio (INR), model of end-stage liver disease (MELD)). * *p* < 0.05. ** *p* < 0.01.

LPC nmol/mL	MELD Score	INR	Albumin g/L	ALT U/L	AST U/L	Creatinine mg/dL	Bilirubin mg/dL
15:0	−0.094	−0.064	0.269 *	−0.046	−0.125	0.024	−0.109
16:0	−0.026	−0.065	0.328 **	0.002	−0.066	0.089	−0.069
16:1	−0.087	−0.047	0.267 *	−0.084	0.003	−0.041	−0.044
18:0	−0.031	−0.062	0.285 *	0.004	−0.116	0.085	−0.057
18:1	−0.011	−0.029	0.304 **	−0.106	−0.062	0.025	−0.004
18:2	−0.090	−0.133	0.183	−0.101	−0.145	0.020	−0.044
18:3	−0.115	−0.117	0.176	−0.103	−0.140	−0.114	−0.040
20:3	−0.119	−0.161	0.266 *	−0.045	−0.167	−0.030	0.026
20:4	−0.038	−0.147	0.233	0.012	−0.142	0.126	0.070
20:5	−0.110	−0.191	0.122	−0.009	−0.158	−0.023	0.100
22:5	0.047	−0.030	0.288 *	−0.123	−0.150	−0.029	0.125
22:6	0.028	−0.029	0.175	0.099	−0.033	0.023	0.147

**Table 5 ijms-25-01198-t005:** Spearman correlation coefficients for the correlation of LPC species and the MELD score, as well as laboratory measures of patients with liver cirrhosis at the end of therapy (alanine aminotransferase (ALT), aspartate aminotransferase (AST), international normalized ratio (INR), model of end-stage liver disease (MELD)). * *p* < 0.05. ** *p* < 0.01, *** *p* < 0.001.

LPC nmol/mL	MELD Score	INR	Albumin g/L	ALT U/L	AST U/L	Creatinine mg/dL	Bilirubin mg/dL
15:0	−0.340	−0.387	0.557 **	−0.278	−0.479 *	0.037	−0.182
16:0	−0.551 **	−0.524 **	0.651 ***	−0.143	−0.404	−0.163	−0.367
16:1	−0.151	−0.118	0.216	−0.096	−0.247	−0.194	0.029
18:0	−0.614 ***	−0.499 *	0.607 ***	−0.083	−0.353	−0.200	−0.466 *
18:1	−0.160	−0.157	0.058	−0.099	−0.150	−0.258	0.008
18:2	−0.336	−0.226	0.249	0.034	−0.345	−0.020	−0.257
18:3	−0.253	−0.172	0.123	0.048	−0.276	−0.147	−0.145
20:3	−0.413	−0.237	0.388	0.065	−0.282	−0.372	−0.339
20:4	−0.601 ***	−0.398	0.586 **	0.007	−0.364	−0.419	−0.464 *
20:5	−0.350	−0.258	0.314	0.088	−0.071	−0.365	−0.234
22:5	−0.348	−0.233	0.393	0.081	−0.225	−0.350	−0.264
22:6	−0.594 **	−0.527 **	0.540 **	0.044	−0.218	−0.263	−0.415

**Table 6 ijms-25-01198-t006:** Area under the receiver operating characteristic curve ± standard deviation for LPC (nmol/mL) and % LPC (LPC species levels relative to total LPC concentration) for discrimination of patients with and without ultrasound-diagnosed liver cirrhosis.

	LPC nmol/mL		% LPC	
	Before Therapy	After Therapy	Before Therapy	After Therapy
15:0	0.263 ± 0.614	0.276 ± 0.640	0.485 ± 0.721	0.534 ± 0.711
16:0	0.197 ± 0.545	0.186 ± 0.545	0.300 ± 0.630	0.408 ± 0.673
16:1	0.377 ± 0.672	0.383 ± 0.683	0.652 ± 0.678	0.682 ± 0.611
18:0	0.235 ± 0.584	0.206 ± 0.570	0.432 ± 0.710	0.409 ± 0.692
18:1	0.390 ± 0.684	0.338 ± 0.640	0.773 ± 0.597	0.720 ± 0.623
18:2	0.342 ± 0.681	0.306 ± 0.636	0.569 ± 0.723	0.528 ± 0.706
18:3	0.386 ± 0.636	0.323 ± 0.678	0.588 ± 0.670	0.505 ± 0.691
20:3	0.230 ± 0.557	0.206 ± 0.539	0.404 ± 0.691	0.350 ± 0.648
20:4	0.200 ± 0.546	0.159 ± 0.477	0.343 ± 0.624	0.293 ± 0.577
20:5	0.182 ± 0.482	0.209 ± 0.575	0.270 ± 0.587	0.324 ± 0.647
22:5	0.260 ± 0.608	0.235 ± 0.590	0.514 ± 0.718	0.486 ± 0.706
22:6	0.139 ± 0.434	0.119 ± 0.373	0.261 ± 0.604	0.231 ± 0.550

## Data Availability

The datasets generated and/or analyzed during the current study are available from the corresponding author on request.

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
