# Peer review of "Rising Lysophosphatidylcholine Levels Post-Hepatitis C Clearance"

_ijms, 2024, doi:10.3390/ijms25021198_

Round 1

Reviewer 1 Report

Comments and Suggestions for Authors

Generally a ell presented manuscript for the most part, the authors may want to consider shading/ hatching for their bar graphs to help with the 2 or 3 bars the same colour.

The authors cite ref 57  for the Lipidomics method, without any further details, this is not good enough, what int stds did they use and importantly how much.?  When were these added? How were the serum samples stored and subsequently  extracted for lipids or Lysol-lipids?  How did they obtain all of the other experimental results in tables 4 and  5 .. nothing about the methods to determine all of these factors.

Table 6 should have ± SD on them and   I am unsure what % LPC in this  table represents.. is it the % on that LPC species wrt total lipid content or Total PC  content probably not ?

Comments on the Quality of English Language

Some punctuation could be improved in several areas  and not starting sentence with "Because"

Author Response

We are very grateful to the reviewer for the very valuable comments, which were of great help to us in the improvement of our manuscript.

Generally a well presented manuscript for the most part, the authors may want to consider shading/ hatching for their bar graphs to help with the 2 or 3 bars the same colour.

This was corrected. This was changes as suggested by the reviewer but this change was not marked in the revised manuscript.

The authors cite ref 57  for the Lipidomics method, without any further details, this is not good enough, what int stds did they use and importantly how much.?  When were these added? How were the serum samples stored and subsequently  extracted for lipids or Lysol-lipids?  How did they obtain all of the other experimental results in tables 4 and  5 .. nothing about the methods to determine all of these factors.

Thank you for this advice. Now these methods are all described in detail. 

Table 6 should have ± SD on them and   I am unsure what % LPC in this  table represents.. is it the % on that LPC species wrt total lipid content or Total PC  content probably not ?

The SDs were included and % LPC is now explained in the table legend. It is the LPC species levels relative to total LPC concentration in %.

Comments on the Quality of English Language

Some punctuation could be improved in several areas  and not starting sentence with "Because"

The two sentences stating with Because were rewritten. We have corrected the text and hopefully found all the mistakes.

Reviewer 2 Report

Comments and Suggestions for Authors

This study aimed to measure serum levels of LPC species 15:0, 16:0, 16:1, 18:0, 18:1, 102 18:2, 18:3, 20:3, 20:4, 20:5, 22:5, and 22:6 in HCV patients before and after DAA therapy. The goal was to investigate the impact of HCV on circulating LPC levels and identify associations with viral load, genotype, and liver disease severity. The subject is well within the scope of the journal, and I personally believe that the research is well organized.

Here are some suggestions for the authors to help improve the quality of the paper.

1. The introduction is too long and the author needs to shorten the introduction, keeping only the sections that are particularly close to the topic and highlight the research gaps.

2. If the statistical methods used to display the data in different tables and figures are different, the authors are advised to specify the methods in the corresponding legend or table example position.

3. I notice that the values of Y for the different metrics in all the figures are different greatly. Therefore, the y-axis of all plots is suggested to be in the form of a component segment, which better shows the differences in the data.

Comments on the Quality of English Language

Minor editing of English language required

Author Response

We are very grateful to the reviewer for the very valuable comments, which helped us to improve our manuscript. We have corrected the figures according to the comment of Reviewer 1 and some bars were hatched for a better differentiation of the different groups and this change is not marked.

This study aimed to measure serum levels of LPC species 15:0, 16:0, 16:1, 18:0, 18:1, 102 18:2, 18:3, 20:3, 20:4, 20:5, 22:5, and 22:6 in HCV patients before and after DAA therapy. The goal was to investigate the impact of HCV on circulating LPC levels and identify associations with viral load, genotype, and liver disease severity. The subject is well within the scope of the journal, and I personally believe that the research is well organized.

Thank you for this kind comment.

Here are some suggestions for the authors to help improve the quality of the paper.

  1. The introduction is too long and the author needs to shorten the introduction, keeping only the sections that are particularly close to the topic and highlight the research gaps.

The introduction was shortened as suggested by the reviewer.

  1. If the statistical methods used to display the data in different tables and figures are different, the authors are advised to specify the methods in the corresponding legend or table example position.

We now mentioned the statistical tests used in the figure legends. The tests used for the data shown in the tables were already included.

  1. I notice that the values of Y for the different metrics in all the figures are different greatly. Therefore, the y-axis of all plots is suggested to be in the form of a component segment, which better shows the differences in the data.
  2.  

The Y axis of the original figures are identical, and the differences are because the figures had different sizes in the word file. We tried to keep comparable sizes in the corrected file.

Comments on the Quality of English Language

Minor editing of English language required

We have corrected the text and hopefully found all the mistakes.